# Reversibly growing crosslinked polymers with programmable sizes and properties

Xiaozhuang Zhou ®[1,2,8], Yijun Zheng[3,8], Haohui Zhang[4], Li Yang[1], Yubo Cui[1], Baiju P. Krishnan[2], Shihua Dong[1], Michael Aizenberg ®[5], Xinhong Xiong[1], Yuhang Hu[4,6], Joanna Aizenberg ®[5,7] ✉ & Jiaxi Cui ®[1,2,5] ✉

Growth constitutes a powerful method to post-modulate materials' structures and functions without compromising their mechanical performance for sustainable use, but the process is irreversible. To address this issue, we here report a growing-degrowing strategy that enables thermosetting materials to either absorb or release components for continuously changing their sizes, shapes, compositions, and a set of properties simultaneously. The strategy is based on the monomer-polymer equilibrium of networks in which supplying or removing small polymerizable components would drive the networks toward expansion or contraction. Using acid-catalyzed equilibration of siloxane as an example, we demonstrate that the size and mechanical properties of the resulting silicone materials can be significantly or finely tuned in both directions of growth and decomposition. The equilibration can be turned off to yield stable products or reactivated again. During the degrowing-growing circle, material structures are selectively varied either uniformly or heterogeneously, by the availability of fillers. Our strategy endows the materials with many appealing capabilities including environment adaptivity, self-healing, and switchability of surface morphologies, shapes, and optical properties. Since monomer-polymer equilibration exists in many polymers, we envision the expansion of the presented strategy to various systems for many applications.

Living organisms are non-equilibrium, open systems that can continuously absorb nutrients from the external environment to grow[1]. During growth, they enlarge[2], change shapes[3], and create fantastic structures and functions[4] to meet different challenges. These processes imply appealing features for materials, and the growth concepts of living organisms have recently been applied to designing artificially intelligent systems[5–14]. Examples range from self-strengthening hydrogels[5] to multicolor structural-color patterns[6]. In these systems,

irreversible polymerizations are typically utilized to convert external nutrients, i.e., monomers and crosslinkers, into the original matrices to tune materials' appearance and properties. This strategy enables many unique properties and features of materials, including the creation of fine textures[7–10] and complex composite structures[6], self-healing of large damages to rigid substrates[11], the evolution of material components and bulk properties[12–14], etc. Despite such progress, the materials' transformations in all these systems are unidirectional due to the

[1]Institute of Fundamental and Frontier Sciences, University of Electronic Science and Technology of China, Chengdu, Sichuan 610054, China. [2]INM - Leibniz Institute for New Materials, Campus D2 2, 66123 Saarbrücken, Germany. [3]School of Physical Science and Technology, ShanghaiTech University, Shanghai 201210, China. [4]The George W. Woodruff School of Mechanical Engineering, Georgia Institute of Technology, Atlanta, GA 30332, US. [5]John A. Paulson School of Engineering and Applied Sciences, Harvard University, Cambridge, MA 02138, USA. [6]The School of Chemical and Biomolecular Engineering, Georgia Institute of Technology, Atlanta, GA 30332, US. [7]Department of Chemistry and Chemical Biology, Harvard University, Cambridge, MA 02138, USA. [8]These authors contributed equally: Xiaozhuang Zhou, Yijun Zheng. ✉e-mail: jaiz@seas.harvard.edu; Jiaxi.Cui@uestc.edu.cn

irreversible nature of the reactions used for nutrient integration, which restricts the flexibility and adaptability of the growth strategy in programming material's structure and properties.

Compared to irreversible reactions, reversible chemical reactions are also common. These reactions are in equilibrium states, in which the addition or removal of starting reactants can drive the systems forward to or reverse from the equilibrium states[15,16]. They usually involve the structures containing reversibly forming covalent bonds[17–19], such as esters[20–24], olefins[25,26], thioesters[27,28], diketoenamines[29,30], and imines[31,32]. While these reactions are widely engaged in thermoset recycling[33–36], we propose transferring their reversible feature from molecular to macroscopic levels to enable three-dimensional materials to grow or degrow. The idea is demonstrated by silicones made via acid-catalyzed ring-opening polymerization[37–40]. In the presence of acidic catalysts, monomer-polymer equilibria form within the materials such that the materials can continuously grow by incorporating supplied substances or degrow by releasing small molecules generated in depolymerization. The composition, shape, and various properties of the materials,

including mechanical performance, surface morphology, and spectroscopic properties, can be selectively tuned through this reversible growth, suggesting unprecedented opportunity in the postmodulation of thermosetting materials.

## Results

### Strategy and systems

Figure 1a shows our design. The materials consist of crosslinked polymers in a catalyst-dependent equilibration (Fig. 1ai). For growth, polymerizable substances, i.e., monomer and crosslinker, are supplied to the systems by swelling (Fig. 1aii) to drive the polymerization toward the formation of new crosslinked networks. Normally, polymer chains of crosslinked networks will be stretched during the swelling, which hinders further uptake of liquid containing molecules of the monomer and crosslinker. In equilibrating polymer networks that are capable of secondary redistribution reactions within the polymer networks, such entropic discomforts will trigger chain exchange between the original and newly formed networks to generate homogenized networks with relaxed polymer conformations (Fig. 1aiii). As a result of the

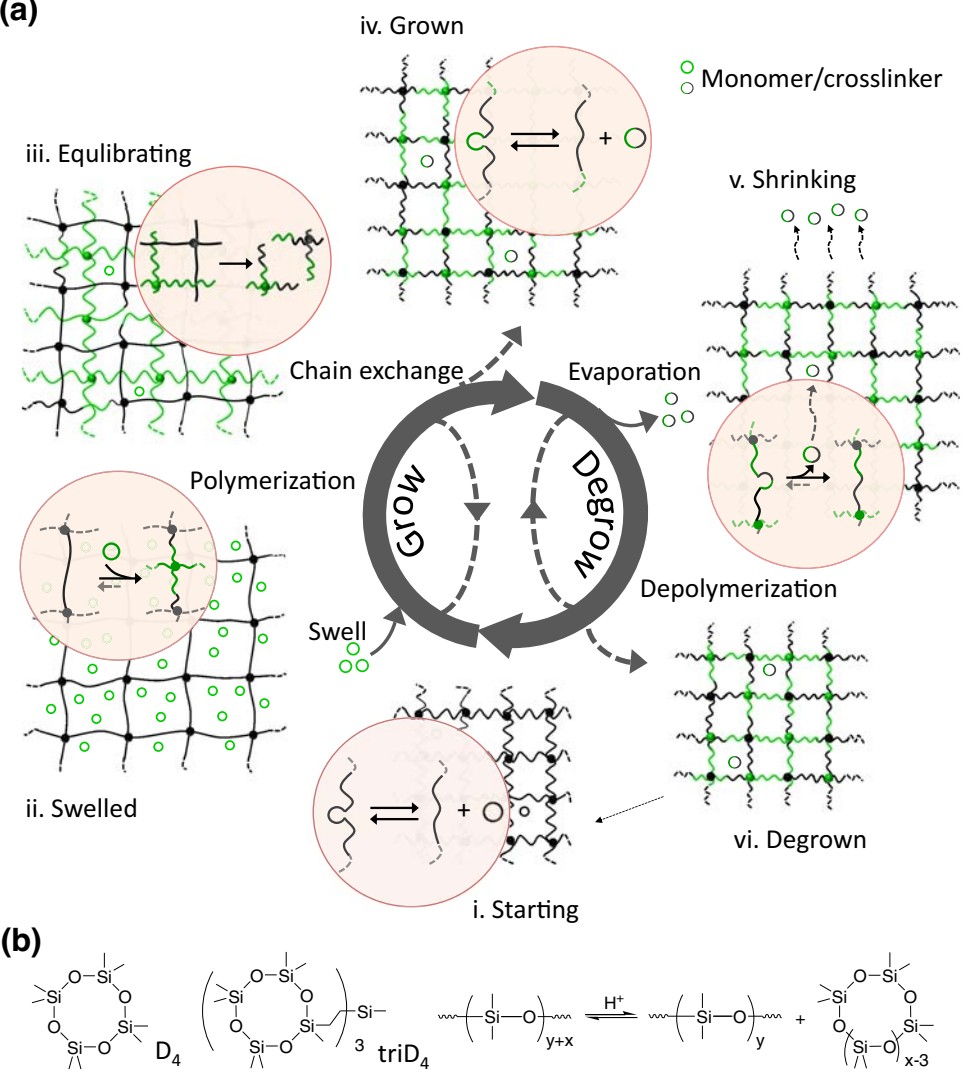

**Fig. 1 | Strategy of enabling crosslinked polymers to grow and degrow. a** Design of the growing and degrowing molecular mechanism of crosslinked polymeric materials. Growing process: a starting sample (**i**) absorbed the mixture of monomer and crosslinker to get the swelled sample (**ii**); polymerization and chain exchange reactions occurred in the equilibrating sample (**iii**) to generate the grown sample

(**iv**). Degrowing process: removal of the monomer by evaporation induced the sample to shrink (**v**) and then generate the degrown product (**vi**). **b** $D_4$ monomer and $triD_4$ crosslinker are used for making living siloxane elastomer and demonstrating the acid-catalyzed monomer–polymer equilibration.

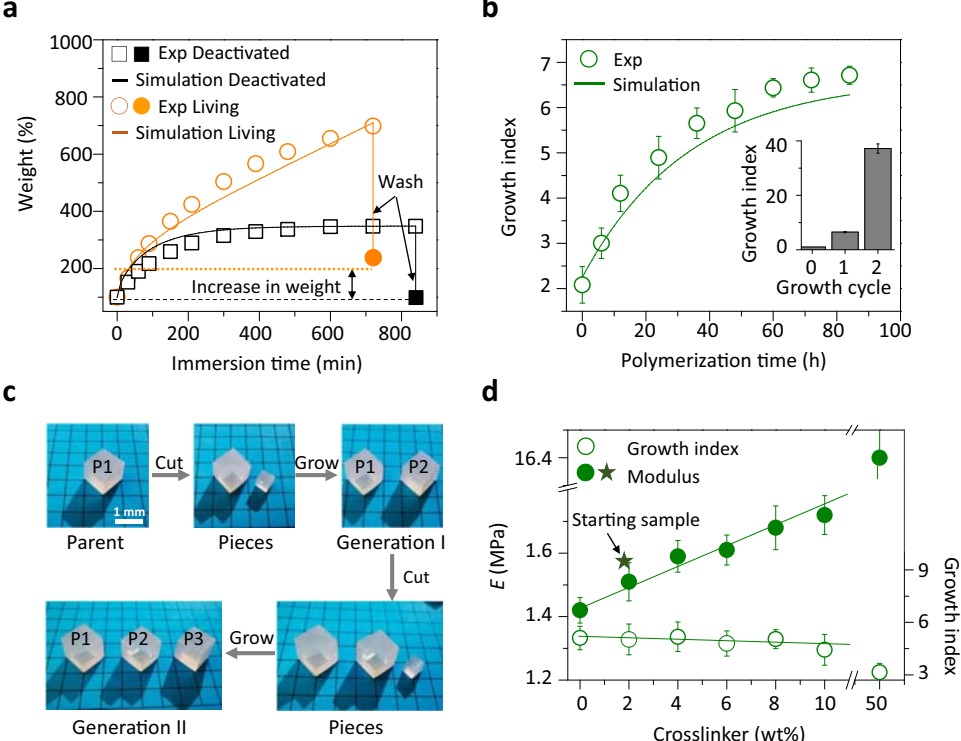

**Fig. 2 | Growth of the living siloxane elastomers. a** Experimental and theoretical time-dependent weight of living and deactivated samples immersed in the monomer and crosslinker mixture solution. Hexane solution containing 1 wt% triethylamine was used to quench the acid-catalyzed reactions and wash out the unreacted monomer and crosslinker. The solid square and circle labels show the weights after washing. **b** Experimental and theoretical growth index of grown products under different conditions. The samples were immersed in the monomer and crosslinker mixture solution for 10 h and then stored in sealed bottles to allow the polymerization and homogenization to progress at different times, followed by hexane washing. A full growth cycle included 10 h immersion and 90 h storage. Growth index: $W_{grown}/W_{original}$. **c** Regeneration of the living samples. A cubic sample (first generation parent, P1) was cut to yield a smaller cubic piece which was grown into a sample (P2) with the same size and shape as P1. The P2 was then used as the second generation parent to regenerate the generation II sample (P3). **d** Young's modulus and growth index of grown products obtained from solutions with different crosslinker concentrations. The mixture solution in **a**–**c** contained 2 wt% triD$_4$. The data in **b** and **d** were obtained from eight independent measurements. Error bars are standard errors of the mean (s.e.m.).

homogenization, the grown and relaxed polymer network can undergo further growth cycles (Fig. 1aiv). Importantly, removing the monomers from the system can drive the polymer networks to shrink (Fig. 1av), leading to degrowth (Fig. 1avi). Based on this equilibrium mechanism, the size, composition, and set of properties of the samples can be tuned finely and reversibly by controlling the availability of polymerizable substances. Besides, the equilibration can be switched off by quenching the catalyst to achieve a stable, desirable product or turned on again by reactivating the catalyst.

Acid-catalyzed equilibrium of siloxane was selected to demonstrate our growth–degrowth concept. The materials were prepared via acid-catalyzed ring-opening copolymerization of octamethylcyclotetrasiloxane (D$_4$) and a star-shaped crosslinker (triD$_4$) with trifluoromethanesulfonic acid as catalyst (Fig. 1b and Supplementary Figs. 1–3). A perylenediimide(PDI)-based dye crosslinker was copolymerized with D$_4$ and triD$_4$ to make samples fluorescent (Supplementary Figs. 4 and 5). The acidic species retained their activity after polymerization and could easily percolate throughout the networks to distribute uniformly and even escape from the sample (Supplementary Fig. 2)[37–42]. To prevent their evaporation and percolation, fillers like carbon black (CB) and silica particles (SiO$_2$) that would adsorb acidic molecules on their surfaces[43] (Supplementary Figs. 6–8) could be selectively added to the materials. The obtained materials were similar to normal silicone elastomers and could swell upon the uptake of the mixture of their parent monomer and crosslinker. During swelling, D$_4$ and triD$_4$ were homogeneously absorbed (Supplementary Figs. 9 and 10 and Supplementary Table 1).

## Growth

To trigger growth, the mixture of monomer and crosslinker was supplied to the equilibrium system by immersing the living samples in a D$_4$ and triD$_4$ mixture solution. The growth could be divided into three overlapping steps: swelling of the reactants to increase the size of samples, in situ polymerization to incorporate these absorbed reactants, and chain exchange to release swelling-induced mechanical tension (Fig. 1a). Since all processes proceeded readily at room temperature (rt), the samples had already undergone growth during the immersion. Figure 2a shows the weight of the samples during the immersion. For comparison, a deactivated sample, in which the acidic species were neutralized to stop the equilibration, was used as a control. The control shows typical swelling kinetics and reaches a stable swelling ratio of 3.45 after 10 h. It completely shrinks back after the absorbed liquid is washed out. In contrast, the weight of the living samples continually increases up to 650 wt% over the same immersion time, and the net weight after washing has more than doubled (around 240 wt%) compared to the starting weight of the sample and that of the control. This extra increase in weight was attributed to the incorporation of the monomer and crosslinker reactants into the polymer chains through the growing cycle. The swelled samples were further stored at rt in a sealed condition to study the kinetics of the system reaching a new monomer-polymer equilibrium (the polymerization was stopped by hexane solution containing triethylamine to remove the monomers and crosslinkers in the sample, which could also be applied to terminate the growth). Triethylamine reacted with triflic acid to form nonvolatile salts[44], which were undissolvable in hexane

and would retain in the polymer matrix during the washing process (Supplementary Fig. 11). Growth index, defined as $W_{grown}/W_{original}$, where $W_{original}$ and $W_{grown}$ are the weights of the starting and grown samples, respectively, was used to describe the kinetics of the equilibration process. The growth index was observed to stabilize after 90 h (Fig. 2b and Supplementary Fig. 12), indicating a remarkably slower (nearly an order of magnitude) polymerization step compared to that of swelling. The faster swelling process allowed the sample to integrate reagents homogeneously. Consequently, the growth was uniform throughout the samples, and the original shape of the samples, though scaled in size, was maintained (Fig. 2c and Supplementary Fig. 13). Notably, the sharp edges displayed negligible change during the growth process, which was evidenced by the close-up photographs and 3D profiles of the sharp edges (Supplementary Fig. 14). We attributed this phenomenon to the consequence of two opposite effects, i.e., chain-exchange reaction and larger surface of the edges. The former smoothed the edges while the latter sharpened the edges.

To facilitate the understanding of the growth mechanism and quantify the growth kinetics, theoretical analyses were carried out. In this system, the polymer chains were constantly changing due to the ongoing chain propagation and exchange reactions between the original and newborn networks. The dynamic chain model for polymers with transient networks was thus adopted and expanded to simulate the growth[45,46]. The model included the following features (see Supplementary 4.5 for more details): (i) chain distribution tensor was used to describe the change in the end-to-end distances of the chains due to polymerization and chain-exchange reaction; (ii) the diffusion of monomers and crosslinkers was described by Fick's law; (iii) the concentration of the catalyst was set to be a constant inside the polymer; and (iv) for the reaction kinetics, the polymerization rate was assumed to be proportional to the monomer and catalyst concentrations, and the chain exchange reaction rate was considered to be proportional to the chain density. The model was calibrated by fitting the theoretical curve of the swelling kinetics and reaction kinetics to the experimental data (black line in Fig. 2a and the solid line in Fig. 2b). Then the model was used to predict the growth kinetics of the living polymer with concurrent swelling and reaction, which compares well with the experimental results as shown in the orange line in Fig. 2a. The results supported the proposed growth mechanism and quantitatively predicted the growth process.

The size of the samples could be finely modulated by controlling either immersion time (which correlates with the amount of supplied monomer and crosslinker) or storage time for a given solution uptake (which correlates with the yield of polymerization due to the slower polymerization kinetics of the swollen samples) in a single growth cycle or significantly magnified by repeating the growth process (inset in Fig. 2b). A growth index of ~40 was observed after two growth cycles when 10 h immersion time was used for each growth cycle. If the sample was immersed in the mixture of monomer and crosslinker (6 wt%) until the sample became too soft to maintain its solid state, a growth index of up to 100 could be achieved. After being stored for seven days, the sample stiffened. This grown sample, as such, was no longer able to directly undergo further growth cycle due to the dilution or quenching of the catalyst. However, when the catalyst was provided to the grown sample, it could undergo growth again. Such feature also implied an interesting regeneration ability: a cubic piece cut from a cubic parent specimen (P1) could be converted into an exact copy of its parent with the same composition and shape (P2), and this process can be repeated indefinitely with the growing offsprings of either the original parent or its descendants (P3) (Fig. 2c).

The composition of the grown samples could be easily tuned by varying the supplied solution, specifically the concentration of the cross-linker, providing a way to modulate their mechanical properties (Fig. 2d). Growth in a solution containing the same crosslinker contents as used for preparing the starting samples led to the same composition and similar mechanical properties while increasing crosslinker concentration in the supplied solution produced stiffer grown samples. In contrast to the significant change in Young's modulus, the growth index did not vary noticeably with the change in the cross-linker concentration in the supplied solution (Supplementary Table 2), suggesting that the overall kinetics of the incorporation of the new material from these solutions remained very similar and fairly independent of the concentration of the crosslinker. The tunable mechanical properties of the grown materials distinguished our strategy from solvent swelling. Simple swelling can expand crosslinked polymer materials, but the change in size is limited by the swelling ratio, and the mechanical properties of the swelled products would decrease significantly.

## Degrowth

In the equilibration of siloxane, depolymerization occurs through backbiting reactions to generate small cyclic molecules[39]. Removal of these depolymerization products from the systems would shift the equilibrium and drive the samples to degrow. The evaporation method was used to remove the volatile small molecules, which were collected by a homemade cold trap for $^1H$ NMR and GC−MS analysis (Supplementary Fig. 15). Analytical results indicated that these volatile products consist of 85.5% $D_4$ and 14.5% $D_5$ (Supplementary Fig. 16, these recycled monomers were pure enough for direct repolymerization to prepare silicone again). Since the starting materials used were $D_4$ and $triD_4$, the formation of other cyclic structures ($D_5$) supported the proposed mechanism of backbiting depolymerization. Figure 3a shows the evaporation-induced degrowth kinetics of the samples under different conditions with a deactivated sample as a control. While the control keeps 99.8% of its weight, filler-free and filler-containing samples shrink by 60% and 85%, respectively, in 180 h. The higher shrinkage in filler-containing systems was attributed to the stabilization by the filler of the otherwise volatile catalyst against evaporation (Supplementary Fig. 17). The dynamic chain model presented here for growth was also used to simulate the degrowth process and its kinetics (Supplementary section 5.2). In this case, the degrowth rate was proportional to the monomer and the catalyst concentrations in the matrix. The theoretical results (solid lines) fit well with the experimental data (Fig. 3a).

The degrowth could be deactivated by neutralizing the acid catalyst to generate stable materials and reactivated again by supplying the acid catalyst to the deactivated systems (Fig. 3b), enabling an "on-off" control. The deactivation can be triggered either by immersing the samples into the triethylamine solution or dropping a small amount of triethylamine on the surface of the samples (the liquid would be absorbed into the matrices). The same control could also be achieved in the growth process. It is noteworthy that the removal of monomer and crosslinker components entrapped in the sample by pure hexane washing or the storage of the sample in a sealing condition to prevent evaporation of small molecules could be employed to stop the growth and degrowth, respectively, but the sample was still in an "on" active state. Growth or degrowth would occur again when a monomer and crosslinker mixture solution was provided to the sample or the sample was exposed to air. In contrast, in "off" states, the grown or degrown samples were stable in the air as normal silicone elastomers, which was demonstrated by the negligible weight loss (<1 wt%) of the deactivated samples after 1-week annealing at 100 °C. We did not expect extra triethylamine residue in the deactivated samples due to its volatile nature, and thus the sample could be activated using the acid catalyst. Fourier-transform infrared spectroscopy was employed to track the triethylamine evaporation from the deactivated samples (Supplementary Fig. 18). After the samples were exposed to air for 2 h, the triethylamine signal significantly decreased, implying the evaporation of extra triethylamine. Besides evaporation, a washing treatment with tetrahydrofuran (THF) could be applied to remove the salts made from

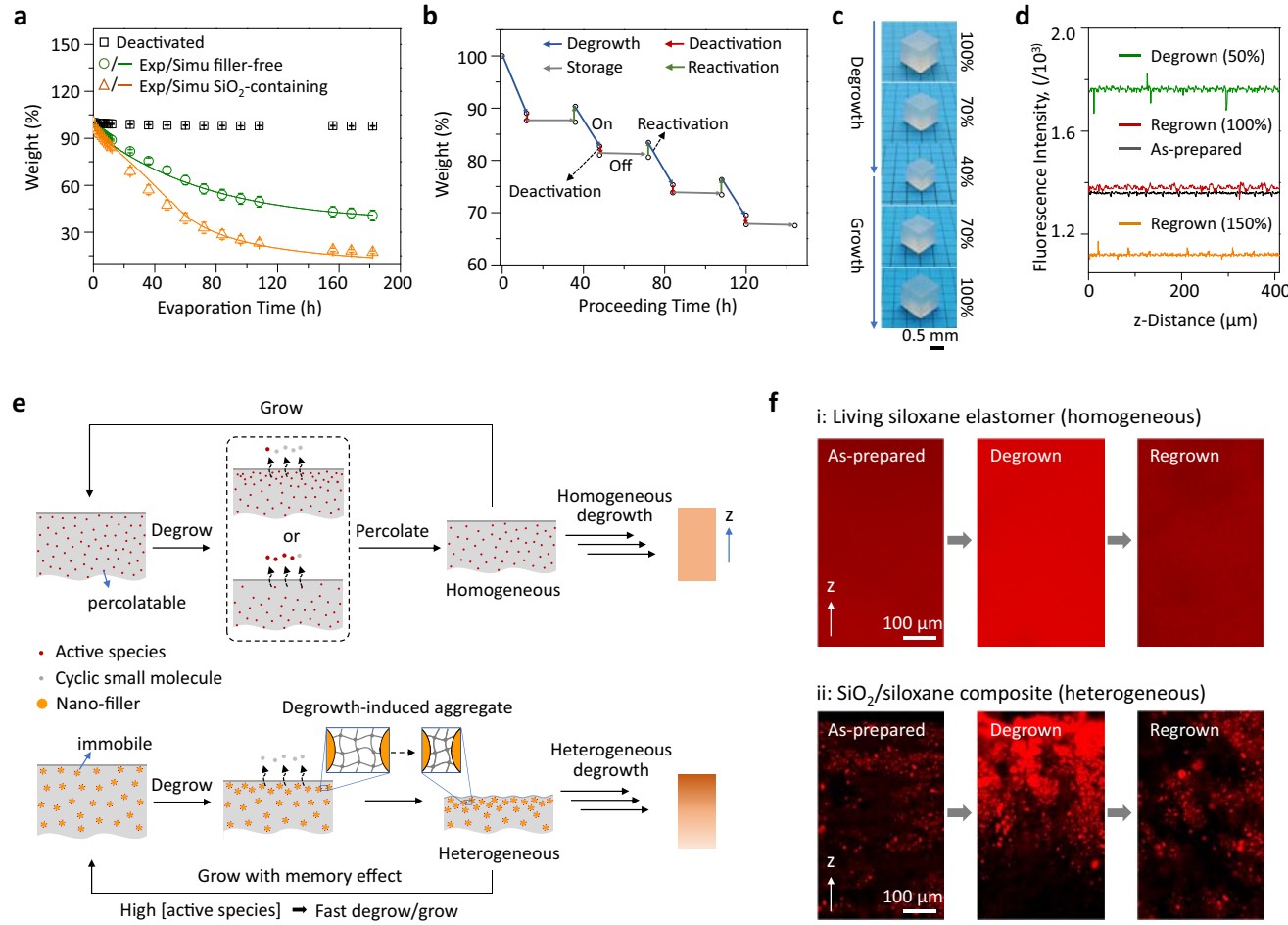

**Fig. 3 | Degrowth of the living siloxane elastomers. a** Time-dependent weight of different samples in the air at rt. Deactivated: deactivated sample; Exp: experimental; Simu: simulated; filler-free: living siloxane elastomer; SiO$_2$-containing: living SiO$_2$/siloxane composite. The experimental data were obtained from eight independent measurements. Error bars are s.e.m. **b** The weight change of a siloxane elastomer in multiple activation-deactivation cycles. The living sample was deactivated by washing the acidic species with triethylamine and re-activated by adding trifluoromethanesulfonic acid. **c** A cubic sample in a degrowth-growth cycle. **d** Fluorescence intensities of a dyed living siloxane elastomer at different states. The as-prepared sample was first degrown to 50% of its initial weight, followed by being regrown to 100% and then to 150% of its initial weight. **e** Schematic mechanisms of homogeneous and heterogeneous degrowth–growth. **f** Vertical cross-section fluorescence images of a dyed living siloxane elastomer (**i**) and a dyed SiO$_2$/siloxane composite (**ii**) at different states.

the acid catalyst and triethylamine generated in the deactivation step. With degrowth, the samples self-stiffened (Supplementary Fig. 19) because the produced crosslinker-like depolymerization products, which were not volatile under the reaction conditions, led to an increase in the degree of crosslinking in the samples.

The degrowth is uniform in the absence of any filler, and the depolymerized smaller products maintain their geometric shapes (Fig. 3c). The samples retain their sharp edges when they lose 40% of their weight. The edges become slightly rough when the weight loss reaches 60% (Supplementary Fig. 20). The degrowth would stiffen the edges, which would favor the maintenance of the sharp edges and might also induce external tension to slightly deform the edges. The process of evaporation-induced degrowth could be divided into two separate steps: (i) depolymerization to generate small cyclic molecules and (ii) the removal of the generated molecules. Since the latter (22 h, Supplementary Fig. 21) is significantly faster than the degrowth (180 h, Fig. 3a), the degrowth rate should depend on the former, i.e., depolymerization. For a sample in which the acidic species triggering depolymerization distributed homogeneously due to the percolation, a homogeneous reduction in size is expected, which should lead to a uniform but the smaller bulk structure of the same shape. To confirm this, we designed a PDI-based dye crosslinker (Supplementary Figs. 4

and 5) whose nonvolatile and crosslinking nature in networks should fully preclude its redistribution after its integration into the network. When a sample dyed with this crosslinker degrew, it displayed intensive and uniform fluorescence (Fig. 3d), implying a homogeneous matrix. After degrowth, the system retained its acidic character and could integrate a monomer and crosslinker mixture solution to grow back to its original state (Fig. 3c, d).

Degrowth was driven by the evaporation of small cyclic products occurring from the surface and inevitably generating a concentration gradient of the active species. In the filler-free systems, such gradient was significantly reduced due to the percolation of the active species (Fig. 3e). Since fillers like CB or SiO$_2$ could immobilize the acidic species, we expected that the gradient could be preserved in the presence of fillers, finally resulting in a heterogeneous degrowth process (Fig. 3e). To prove this hypothesis, we investigated the bulk structure of degrown dyed SiO$_2$/living siloxane composites with a filler-free sample as a control (Fig. 3f). The control shows uniform matrices in the as-prepared, degrown and regrown states due to the homogenization of percolating species in the sample. In the SiO$_2$/siloxane composites, the dye-functionalized crosslinker is readily adsorbed on the acidic SiO$_2$ through the imide(PDI)–acid interaction to make the SiO$_2$ particles brightly fluorescent (Fig. 3fii and Supplementary Figs. 22 and 23).

These bright particles distribute homogeneously in the as-prepared sample but form a gradient in the degrown sample, i.e., more fillers in the outer region, suggesting faster degrowth in the exterior. To further confirm the mechanism, composites made from hydrophobic polytetrafluoroethylene particles that could not immobilize acidic species were subjected to the same treatment. In these composites, matrices rather than particles are brightly fluorescent and remain homogeneous throughout different states (Supplementary Fig. 24). Interestingly, a homogeneous structure formed again when the degrown SiO₂/siloxane composite was regrown, implying a structure–memory effect in the heterogeneous degrowth-growth cycle (Fig. 3fii). We attributed the structure-memory effect to the relative immobility of the particles in the matrices due to their big sizes. These fixed particles allowed the regions showing faster depolymerization in degrowth to undergo similar faster polymerization in growth (Fig. 3e). To confirm this mechanism, we compared the growth rates of the degrown samples with different shrinking ratios (10%, 15%, and 20%). The degrown sample with higher shrinking ratios (higher concentration of acid-adsorbing SiO₂) showed faster growth when they were grown in the same mixture solution (Supplementary Fig. 25).

### Controlling the material properties of grown–degrown samples

Several experiments were conducted to demonstrate the potential of the growing–degrowing strategy in post-manipulation of the material's size, shape, and various material properties (Fig. 4 and Supplementary Sections 6.1–6.5). (1) By taking advantage of homogeneous growth, a series of scaled replicas of surface micro-patterns with programmable sizes could be prepared from a single substrate (Fig. 4a). When the microstructures were magnified, their aspect ratio was well maintained. (2) In contrast to the uniform growth, the growth of the samples also showed incredible adaptability to geometric restriction. As shown in Fig. 4b, two small dyed pieces placed in two corners of an H-shape chamber, respectively, would grow along the wall when a dye-free mixture solution was supplied. During the growth, the samples indeed continually reshaped themselves to adapt to the mold and finally joined together, resulting in an integrated H-shape elastomer. The integrated product would show uniform fluorescence throughout the sample under irradiation, indicating that the dye crosslinkers in pieces have been homogeneously redistributed during the growth. (3) Importantly, the final seamless merging of the two separately growing parts into a uniform H-shape also suggested an effective self-healing ability. Indeed, further tests indicated that all the as-prepared, degrown, and regrown samples display excellent self-healing ability at rt without any external stimulus due to the ongoing rearrangements of siloxane networks (Fig. 4c). All the samples could self-heal from the pieces that had been completely cleaved, and the healed samples exhibit nearly the same stress-strain curves as the intact samples. (4) The heterogeneous degrowth-growth could be utilized to switch surface morphologies. For example, when a SiO₂/siloxane film with a translucent and smooth surface was allowed to degrow to expose the undegradable fillers, an opaque rough coating was obtained (Supplementary Fig. 26). Since the fillers were still coated with siloxane (Supplementary Fig. 26), the sample switches from hydrophobic to superhydrophobic (Fig. 4d, top row). (5) Moreover, the superhydrophobicity could be patterned on the surface by using masks to control localized molecular evaporation resulting in the creation of regionally degrown superhydrophobic patches (see Schematic in Fig. 4d and Supplementary Fig. 27). The patterns could be erased by growth and rewritten again. Stable patterns could be obtained by deactivation treatment (Supplementary Fig. 28). (6) The shrinkage in degrowth could also be harnessed to create actuatable materials (Supplementary Fig. 29), which was demonstrated in controllable deformation of the degrown and regrown samples (Fig. 4e). Such deformation was also reversible in the heterogeneous degrowth–growth cycle.

## Discussion

We have developed a new strategy for enabling thermoset materials to continuously modulate their sizes, shapes, compositions, and set of properties on demand. The strategy is based on the expression of monomer-polymer equilibration from molecular to macroscopic material levels. By supplying or removing polymerizable substances to shift the equilibria, the materials were induced to grow or degrow. The growth or degrowth can be reversibly turned off to yield stable materials. With the availability of fillers, the degrowing–growing processes can be selected to be uniform or heterogeneous. Taking full advantage of these mechanisms, we have demonstrated many appealing features of the resulting materials, including regeneration ability, environment adaptivity, self-healing ability, and switchability in surface morphologies, optical properties, and shapes. The acid-catalyzed siloxane polymerization/depolymerization demonstrated here is only one representative example of many reversible processes that can occur in polymer materials, even though it is a widely used material for various applications, which suggests many immediate, practically important opportunities. Besides post-modulating materials for more sustainable use, reversible growth also represents a new attractive method for material fabrication in which waste generation is minimized. We envision this new paradigm to have significant potential in smart materials for meeting many emerging challenges, like the reuse of thermoset materials.

## Methods

### Materials

Octamethylcyclotetrasiloxane (D₄, 98%, Sigma-Aldrich), trifluoromethanesulfonic acid (triflic acid, 99%, Sigma-Aldrich), heptamethylcyclotetrasiloxane (tech-95, Gelest), trivinylmethylsilane (95%, Gelest), platinum(0)-1,3-divinyl-1,1,3,3-tetramethyldisiloxane in xylene (Pt catalyst, 2% of Pt, Sigma-Aldrich), perylene-3,4,9,10-tetracarboxylic dianhydride (97%, Sigma-Aldrich), and solvents were used as purchased. CB pearls 2000 (BP-2000) were offered by Cabot Corporation, silica nanoparticles (20 nm) were purchased from Sigma-Aldrich, and PTFE nanoparticles (MPD170, 20-50 nm) were purchased from DuPont. All the chemicals were used directly without further treatment.

### Instruments

Solution ¹H spectra were measured in CDCl₃ solution at 25 °C using Varian M400 400 MHz or I500C 500 MHz spectrometer. UV spectra were measured on Agilent 8453 UV-Vis spectrometer. Fluorescence images were obtained by using a fluorescence confocal microscope LSM 710 (ZEISS). The mechanical properties were tested on an Instron Model 5566 (Instron). Gas chromatography–mass spectrometry (GC–MS) was performed on GC–MS Shimadzu QP 2010 using ZB-5HT-Inferno columns. Scanning electron microscopy (SEM) images and energy-dispersive X-ray (EDX) elementary mapping were performed on an FEI Quanta 400 FEG-ESEM with an operating voltage of 3.0 kV. The contact angle measurements were conducted using an OCA 50 AF (DataPhysics). A sessile water drop (5 μl) was dispensed onto the sample surfaces, and the contact angles were determined by the provided software (SCA 20). The 3D morphology of the sample surface was characterized by the 3D Profilometer (NANOVEA ST400, USA). An ultrasonic cleaner (97043-996-EA, VWR) was used to disperse the particles in the liquid solution.

### Preparation of living siloxane elastomers

To the solution, (2–20 ml) of D₄ and triD₄ was added triflic acid. A highly viscous liquid formed immediately. After 5 min, an elastomer formed, which was sealed and stored at rt for 12 h before use. Typically, the starting samples for the growth were made from a D₄ solution containing 1 or 2 wt% triD₄. The samples for the degrowth normally contained 1 wt% triD₄ and 1 wt% triflic acid. To prepare dyed samples,

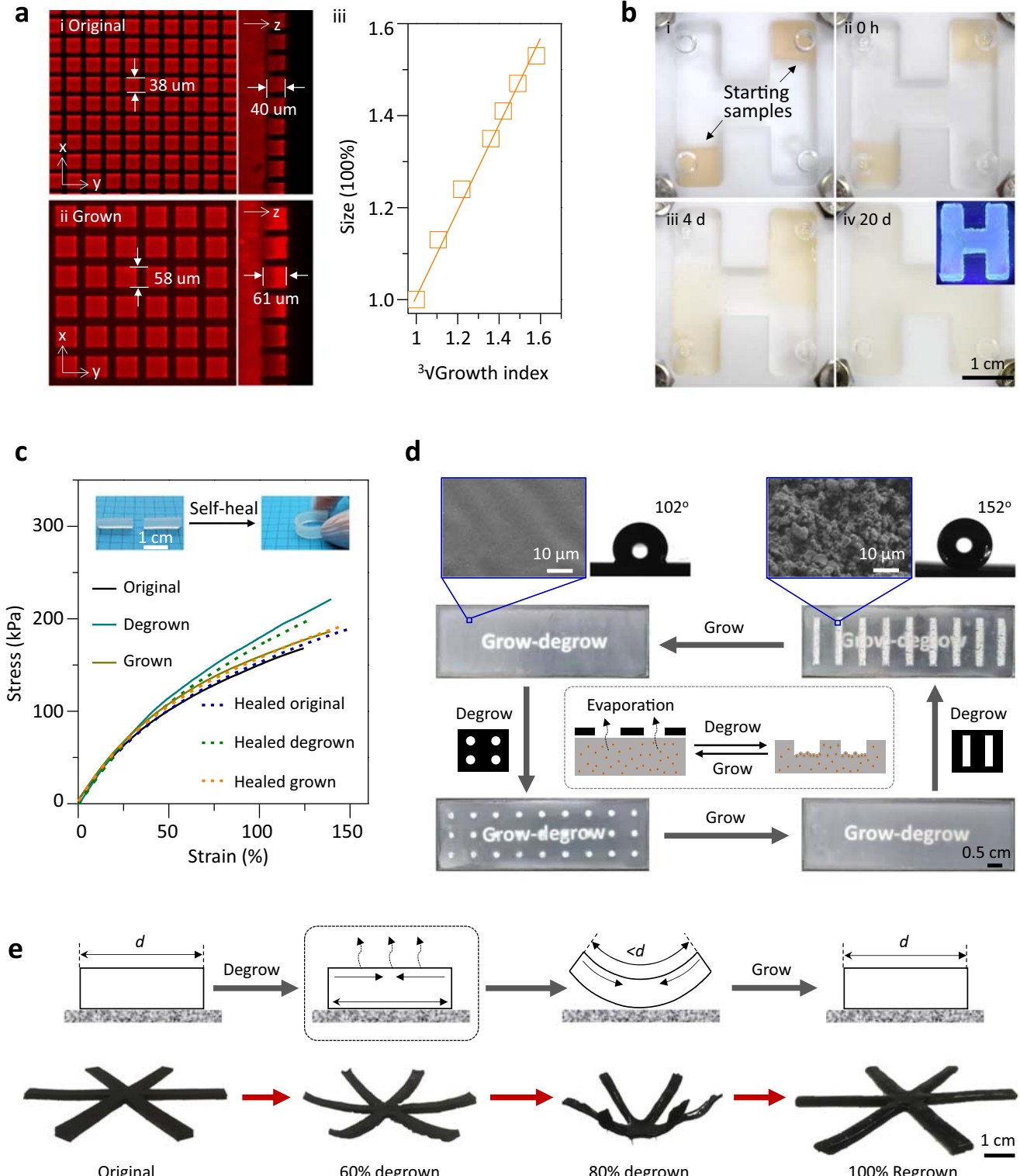

**Fig. 4 | Structure modulation by growth and degrowth. a** Growth of dyed microposts made from living siloxane elastomer: fluorescence image (left: horizontal cross-section; right: vertical cross-section) of the original sample (**i**) and its grown product (**ii**); (**iii**) Plot of the size of the microposts against the cubic root of the growth index. **b** Confinement growth of two dyed samples (**i**) in a chamber for different times (**ii**–**iv**): h: hour; d: day. The dye-free mixture solution was used. Inset: the final product showed homogeneous emission under UV irradiation.
**c** Demonstration of self-healing and the typical tensile stress-strain curves of different samples. Original: as-prepared sample; Degrown: degrown sample (70% of the initial weight); Grown: grown sample (with a grown index of 1.3). The healed samples were obtained by putting two pieces of the samples together and storing overnight in a sealed condition. Typical uniaxial tensile tests with a loading rate of 20 mm/min were conducted to get the stress−strain curves. **d** Reversible patterning of a SiO$_2$/siloxane film. The surface was observed by SEM, and the water contact angles were obtained from a big enough non-patterned sample. **e** Deformation of a CB/siloxane sample in degrowth–growth. The degrowth of the CB/siloxane sample was conducted by placing the sample on the petri dish and allowing one of its surfaces to be exposed to the air.

the PDI-based dye crosslinker (0.02 wt%) was mixed with $D_4$ and $triD_4$ before the addition of triflic acid.

## Preparation of filler-containing siloxane elastomers

To prepare filler/living siloxane composites, fillers were first dispersed in the mixture of $D_4$ and $triD_4$ (1 wt%) by sonification for 30 min, followed by the addition of triflic acid (1 wt%). The mixture was then stored in a sealed container for 12 h.

## Deactivation of (filler-containing) living siloxane elastomers

As-prepared samples were immersed in triethylamine for 4 h, and then the absorbed triethylamine was removed by putting the samples in a hood for solvent evaporation. After three swelling-drying cycles, the living siloxane elastomers were fully deactivated. Another way to deactivate the as-prepared samples was to coat magnesia powder (MgO) on the surface of the samples and to keep them in contact for 10–15 hours (typically overnight). This method was reported previously[39]. The effectiveness of this deactivation method indicates that the active species can easily transfer throughout the sample and supports the mechanism described in Supplementary Fig. 2.

The deactivation of filler-containing siloxane elastomers was performed by immersing the samples in triethylamine. After being immersed for 2 h, the samples were dried in a hood at rt. This process was repeated three times.

## Data availability

The authors declare that the main data supporting the findings of this study are available within the article and its Supplementary Information files. Extra data are available from the corresponding author upon request.

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

## Acknowledgements

The work was originated and supported by the US Department of Energy (DOE), Office of Science, Basic Energy Sciences (BES) under award number DE-SC0005247, and was further financially supported by the National Natural Science Foundation of China (51973023, 52073175, 52003035), Sichuan Science and Technology Program (2021JDRC0014 and 2021JDRC0106), and Shanghai Pujiang Program (20PJ1411200).

## Author contributions

X.Z. and Y.Z. equally contributed to this work. J.C., J.A., and Y.Z. conceived the concept. J.C. and J.A. supervised the project. X.Z., Y.Z., J.C., L.Y., Y.C., B.K., S.D., and X.X. conducted the experiments. H.Z. and Y.H. conducted the simulation. M.A. analyzed data. J.C., X.Z., Y.Z., H.Z., Y.H., M.A., and J.A. wrote the paper. All authors contributed to the analysis and discussion of the data.

## Competing interests

The authors declare no competing interests.
