## [Peer Review File · Nature Communications]

Reversibly growing crosslinked polymers with programmable sizes and propertiesReviewers' Comments:

Reviewer #2:

Remarks to the Author:

The technical comments, including those of reviewer 2, have been addressed sufficiently. Pin-pointing the relevance of this work and how it relates to the state-of-the-art is still challenging. The authors' response to the corresponding comments of all three reviewers remains somewhat evasive. More importantly, this also remains unclear in the manuscript. The reference to growth of living organisms, given in the introduction as starting point and motivation of this work, is far-fetched. Beyond the superficial analogy of an object increasing in size there is no resemblance, e.g. no metabolism is involved in the system describe here. This does not help to recognize the relevance and advances made here.

Reviewer #3:

Remarks to the Author:

Zhou and colleagues have submitted a revised version of their manuscript to Nature Communications, detailing their findings on the growth and degrowth of thermosetted siloxane materials. Their work presents an exciting new concept for customizing the shape and properties of crosslinked polymeric materials as needed, a feature which is often deemed challenging, if not impossible. This innovative approach is poised to have a significant impact on the development of smart and reprogrammable materials.

The authors addressed the comments of the three reviewers clearly and point-by-point and made substantial changes to the manuscript. However, there is one aspect that, in my opinion, needs further attention before the manuscript can be considered to be published in Nature Communications.

The authors utilized triethylamine as a base to neutralize and quench the trifluoromethanesulfonic acid. However, a potential limitation of this approach is the retention of the generated salt within the material even after thorough washing. The presence of this salt can significantly alter the material's properties, such as its polarity and affinity for water. Moreover, the accumulation of salt after each quenching step (as depicted in Figure 3B) may exacerbate its impact over time. It would be advantageous to identify a suitable solvent capable of removing the salt or explore alternative methods for its removal after quenching. In the event that this is not feasible, the authors should conduct a comprehensive analysis of the impact of the remaining salt on the material's long-term stability, water uptake over time, potential phase segregation, or other relevant parameters. Although the formation of the salt may seem like a trivial detail, its presence can substantially alter the material's properties. Hence, a detailed discussion on this aspect would significantly bolster this programming concept.

REVIEWERS' COMMENTS

Reviewer #2 (Remarks to the Author):

The technical comments, including those of reviewer 2, have been addressed sufficiently.

Pin-pointing the relevance of this work and how it relates to the state-of-the-art is still challenging. The authors' response to the corresponding comments of all three reviewers remains somewhat evasive. More importantly, this also remains unclear in the manuscript. The reference to growth of living organisms, given in the introduction as starting point and motivation of this work, is far-fetched. Beyond the superficial analogy of an object increasing in size there is no resemblance, e.g. no metabolism is involved in the system describe here. This does not help to recognize the relevance and advances made here.

Reviewer #3 (Remarks to the Author):

Zhou and colleagues have submitted a revised version of their manuscript to Nature Communications, detailing their findings on the growth and degrowth of thermosetted siloxane materials. Their work presents an exciting new concept for customizing the shape and properties of crosslinked polymeric materials as needed, a feature which is often deemed challenging, if not impossible. This innovative approach is poised to have a significant impact on the development of smart and reprogrammable materials.

The authors addressed the comments of the three reviewers clearly and point-by-point and made substantial changes to the manuscript. However, there is one aspect that, in my opinion, needs further attention before the manuscript can be considered to be published in Nature Communications.

The authors utilized triethylamine as a base to neutralize and quench the trifluoromethanesulfonic acid. However, a potential limitation of this approach is the retention of the generated salt within the material even after thorough washing. The presence of this salt can significantly alter the material's properties, such as its polarity and affinity for water. Moreover, the accumulation of salt after each quenching step (as depicted in Figure 3B) may exacerbate its impact over time. It would be advantageous to identify a suitable solvent capable of removing the salt or explore alternative methods for its removal after quenching. In the event that this is not feasible, the authors should conduct a comprehensive analysis of the impact of the remaining salt on the material's long-term stability, water uptake over time, potential phase segregation, or other relevant parameters. Although the formation of the salt may seem like a trivial detail, its presence can substantially alter the

material's properties. Hence, a detailed discussion on this aspect would significantly bolster this programming concept.

Reply: Thanks for the insightful comments! We have conducted additional experiment to prove that the formed salts can be removed by a simple washing treatment with tetrahydrofuran (THF) as the solvent. As shown in **Figure R1**, the peak at around 1226 cm^{-1} is assigned to C-F stretching of salts, which is gone after THF washing.

We have added the sentence in the main text.

“Besides evaporation, a washing treatment with tetrahydrofuran (THF) could be applied to remove the salts made from the acid catalyst and triethylamine generated in the deactivation step.”

We have also described the detailed measurement in the Supplementary Information 3.10 “The residual salts within the deactivated samples could be removed by washing with tetrahydrofuran (THF). Briefly, the samples were immersed in THF for 30 minutes, followed by drying in a fume hood for 12 hours for solvent evaporation. This process was repeated three times to ensure thorough removal of the salts. ATR-FTIR spectroscopy was used to monitor the washing process. As shown in Supplementary Figure 14b, the as-prepared deactivated samples showed a characteristic peak at around 1226 cm^{-1} , which was attributed to the C-F stretching of the salts. After washing, this peak disappears, indicating that the salts could be effectively removed from the samples.”

Figure R1. (a) ATR-FTIR spectra of the living siloxane, triethylamine, deactivated siloxane at different evaporation times. (b) ATR-FTIR spectrum of the THF washed deactivated siloxane.